# Therapeutic Efficacy of Palmitoylethanolamide and Its New Formulations in Synergy with Different Antioxidant Molecules Present in Diets

**DOI:** 10.3390/nu11092175

**Published:** 2019-09-11

**Authors:** Alessio Filippo Peritore, Rosalba Siracusa, Rosalia Crupi, Salvatore Cuzzocrea

**Affiliations:** 1Department of Chemical Biological, Pharmaceutical and Environmental Sciences, University of Messina, Viale Ferdinando Stagno D’Alcontres 31, 98166 Messina, Italy; 2Department of Pharmacological and Physiological Sciences, Saint Louis University School of Medicine, St. Louis, MO 63104, USA

**Keywords:** Palmitoylethanolamide, ALIAmide, NAcylethanolamide

## Abstract

The use of a complete nutritional approach seems increasingly promising to combat chronic inflammation. The choice of healthy sources of carbohydrates, fats, and proteins, associated with regular physical activity and avoidance of smoking is essential to fight the war against chronic diseases. At the base of the analgesic, anti-inflammatory, or antioxidant action of the diets, there are numerous molecules, among which some of a lipidic nature very active in the inflammatory pathway. One class of molecules found in diets with anti-inflammatory actions are ALIAmides. Among all, one is particularly known for its ability to counteract the inflammatory cascade, the Palmitoylethanolamide (PEA). PEA is a molecular that is present in nature, in numerous foods, and is endogenously produced by our body, which acts as a balancer of inflammatory processes, also known as endocannabionoid-like. PEA is often used in the treatment of both acute and chronic inflammatory pathologies, either alone or in association with other molecules with properties, such as antioxidants or analgesics. This review aims to illustrate an overview of the different diets that are involved in the process of opposition to the inflammatory cascade, focusing on capacity of PEA and new formulations in synergy with other molecules.

## 1. Introduction

Inflammation itself is not a negative mechanism. Indeed, it is the first biological response of our organism to an external aggression. It is functional to the state of health when it is aimed at a specific biological event, when it has a start time and of precise resolution, when the intensity is limited, and when the pain that accompanies it does not interfere with the quality of everyday life. Inflammatory processes are usually triggered by an attack by physical agents (such as an injury, trauma, or burn), toxic, chemical, or biological (such as bacteria and viruses). The duration of the inflammation depends on the time that is needed to eliminate the harmful cause and to repair the damage. It can be short (hours or days) or it can last months or years. In the second case, the inflammatory processes, from defensive become a harmful mechanism [1]. Inflammation becomes pathological when an imbalance is created in the relationship between pro-inflammatory and anti-inflammatory mediators in a sort of adaptive mechanism to a person’s permanent exposure to stress factors, so the inflammation continuously triggers anti-inflammatory responses. Therefore, chronic inflammation, which lasts over time, causes several problems to both affected organs and those (“comorbidity”) [2]. It is widely accepted that old age is characterized by a low-grade and chronic inflammatory state, which has been called inflammaging [3,4,5]. In addition, it is believed that inflammation is at the beginning of several diseases, including obesity, cancer, neurodegenerative, and cardiovascular diseases [6]. Ultimately, the percentage of people suffering from inflammation and pain of various kinds is very high. Sometimes, these conditions are so extreme that they are disabling. For this reason, and for rapid relief, synthetic anti-inflammatory drugs (drugs) are often used. Treatment with the emergency drug may be useful, but people who use this solution usually do so in a recurrent and routine manner causing addiction and worsening the general condition of the body. Therefore, it is always good to make prevention your weapon to avoid resorting to solutions that in the immediate will relieve us from the inflammation, but that then, on the other hand, will damage us even more [7,8]. In this context, a good diet can offer an excellent alternative to the traditional drug. In fact, there are foods that have some wonderful anti-inflammatory and antioxidant characteristics. Palmitoylethanolamide (PEA) is both a naturally occurring lipid ingredient contained in foods/dietary supplements and an endogenous lipid mediator belonging to the class of fatty acid ethanolamides [9]. Evidence indicates that PEA is an important anti-inflammatory, analgesic, and neuroprotective mediator acting on several molecular targets in both central and in peripheral organs and systems [10]. However, PEA lacks a direct antioxidant capacity to prevent the formation of free radicals, and to counteract the damage of DNA, lipids, and proteins. Therefore, in this review, we will see how the PEA in synergy with the natural antioxidant molecules that are present in some foods can have positive therapeutic effects, both on inflammatory processes and on oxidative stress.

## 2. *N-*Acylethanolamines and Diet

A class of important component that can be obtained through the diet are polyunsaturated fatty acids (PUFA), which play fundamental roles in numerous cellular processes, such as inflammation, immunity, or even neurotransmission. With the diet, it is essential to maintain a correct balance between the omega-6 (ω-6) PUFA and the PUFA ω-3. The typical Western diet contains an excess of ω-6 and at the same time a deficiency of ω-3s [11]. Arachidonic acid (AA) is the archetype ω-6, with 20 carbon atoms and four double bonds (20: 4ω-6). In western populations, there are numerous chronic diseases due to AA derivatives, such as pain and swelling that are caused by prostaglandins and bronchoconstriction and asthma by leukotrienes. PUFA ω-3 is intended to counteract the inflammatory metabolites of AA. Eicosapentaenoic acid (EPA, 20: 5ω-3) and docosahexaenoic acid (DHA, 22: 6ω-3) are two of the most well-known PUFA ω-3 that are able to counter the inflammatory metabolites of AA. Other AA derivatives are also some *N-*Acylethanolamines (NAEs), which are known molecules for their anti-inflammatory action. Numerous preclinical studies have shown that a massive integration of diet rich of AA causes the increase in serum levels of two of the best known NAEs, Anandamide (AEA) and 2-Arachidonoylglycerol (2-AG). The need for AA intake for the biosynthesis of the NAEs can therefore lead to a massive increase in the latter’s levels, and in the long run it can lead to a desensitization of the Cannabinoid receptor 1 (CB1) and a downregulation of the Cannabinoid receptor 2 (CB2) [12,13,14]. Linoleic acid, a PUFA 18: 2ω-6, is converted to AA, increase 2-AG and AEA levels, and induces obesity in mice [15]. The use of the dietary supplement that is based on ω-3s leads to an increase in the concentration of EPA and/or DHA in tissues, cells, and plasma, and at the same time a decrease in the concentration of AA in the same [16,17]. A diet that is rich in ω-3 markedly elevated the NAE metabolite of DHA, called DHEA, the NAE metabolite of EPA, called EPEA, and the sn-2 glycerol-ester metabolite of EPA, called 2-EPG [12,17]. DHEA and EPEA showed a high binding affinity for CB1 (Ki = 124 and 55 nM respectively) and acted as partial agonists, act like the endocannabinoids, and also DHEA is catabolized by fatty acid amide hydrolase (FAAH) [18]. Their affinity is almost the same as that of the AEA [19]. In natural fish oil, DHA and EPA are esterified in triacylglycerides (TAGs), while in many fish oil capsules, DHA and EPA are esterified into EE (ethyl ester) or TAG (rTAG). Krill oil contains both DHA and EPA, which are esterified in phospholipids, mainly phosphatidylcholine, which can improve their bioavailability; in addition, krill oil contains less AA than fish oil [20]. In another study it has been seen that the integration of milk formulas with AA acid and docosahexaenoic acid (C22: 6n3, DHA) increases the concentrations of the corresponding NAE, AEA, and *N-*docosahexaenoyl-ethanolamine (22: 6n3 NAE) in specific brain regions of newborn piglets, Ref. [12], while 2-AG levels have not been changed. In the same study, mice that were fed an AA-rich diet from the post-natal day 1-58 exhibited an increase in AEA concentrations in the whole brain [12]. In a study [13], see that after a diet deficient of ω-3 PUFA, mice show higher level of 2-AG in the brain, while in another study a decrease of 2-AG brain levels can be seen after short-term supplementation of DHA-rich fish oil (FO) as compared with the diet supplemented with low n-3 PUFA. It has been found that the presence in the diet of some unusual fatty acids, such as conjugated linoleic acid (CLA), is linked with changes in the cerebral endocannabinoid levels. In fact, in 3-wk-old mice, they were fed diets containing 3% LA or CLA for four weeks. The amount of 2-AG, but not anandamide, OEA and PEA, in the cerebral cortex, but not the hypothalamus, was significantly decreased by treatment with CLA as compared to LA [21]. Furthermore, the authors of this study used LA, the biosynthetic precursor of AA, as a control fatty acid and, as a consequence, the CLA diet was 3% lower than LA when compared to controlling the diet, thus probably influencing both levels of AA and 2-AG of the brain, also independently of dietary CLA. In the summary, the possible effects of CLA on endocannabinoids should be studied while using lower doses and another adequate control diet.

## 3. ALIAmides, PEA and Food

One family of molecules involved in neuroinflammation is the ALIAmides family. The acronym ALIAmides derives from the definition Autacoid Local Injury Antagonist amides (ALIAmides) coined for the first time by the group of Rita Levi Montalcini describing a group of endogenous bioactive acyl ethanolamides with anti-inflammatory properties, such as the renowned palmitoyl ethanolamide (PEA), or stearoyl ethanolamide (SEA) and oleoyl ethanolamide (OEA) [22]. *N-*(2-hydroxyethyl)-exadecanamide, commonly known as palmitoylethanolamide (PEA), belongs to the *N-*acylethanolamine family (NAE). During a study that was conducted in 1950, some researchers found that patients, in that case less privileged children, which were subjected to a diet rich of dry chicken and egg yolk did not contract rheumatic fever. Looking further into the study, they noted that the purified lipid fraction of egg yolk was responsible for the anti-inflammatory action [23,24]. Only in 1957 did they understand that it was the PEA that exerted the anti-inflammatory action, present not only in egg yolk, but also in other foods, such as soy oil, peanut oil, and corn. The potential therapeutic use of PEA has led many researchers to identify other natural sources that are rich in this compound. In fact, PEA has also been found in the seeds of some varieties of legumes, such as peas and beans [25,26], as well as in some varieties of vegetables, such as tomatoes and potatoes [25,27]. Finally, high levels of PEA were also found in human, bovine, and moose milk [28,29]. PEA, like endocannabinoids, is an endogenous lipid molecule 25 that is produced “on demand” by most of the body’s cells and tissues, and it acts as a balancer in those processes that are associated with neuro-inflammation [10,30,31].

## 4. Diet Modify PEA Levels

A different treatment strategy might consider the function of diets on the endogenous levels of PEA, as PEA is the ester between palmitic acid and ethanolamine. Probably, increasing the intake of one of these two compounds through the diet could increase the levels of the PEA itself. Our body produces endogenous palmitic acid, but at the same time it can be found in different dates. Various foods possess palmitic acid, such as foods that are rich in animal fat triglycerides, various vegetable oils, such as cottonseed oil or palm oil. It was shown that no effect on the three groups of fatty acids, the dietary saturated fatty acids, the monounsaturated fatty acids, and the polyunsaturated fatty acids, in membrane phospholipids after extensive changes in the ratio of these three groups, but at the same time considerable influence on fatty acid composition of adipose tissue and plasma triglycerides [32]. No effect on PEA levels in the brain and liver, as well as in the percentage of palmitic acid in fatty acid composition of bulk phospholipids of the same two tissues, was shown after one week feed rats with high-fat-diet, enriched with 9.3-fold with palmitic acid [33]. Dietary fat was found to decrease the levels of PEA in the small intestine (jejunum), furthermore after fed with palmitic acid-rich palm oil, the PEA levels in jejunum did also decrease, but the PEA levels decrease was little less than other NAEs. In another study, it was observed [34] that high dietary fat for eight weeks decreased the PEA levels in the small intestine of mice, but not after 14 weeks of high-fat diet. To date, no data exists regarding the effect of dietary ethanolamine on PEA or other NAE tissue levels, although it has been seen that the contribution of an ethanolamine supplement increases the levels in the brain of phosphatidylethanolamine and also in liver microsomes [35,36]

## 5. Biosynthesis, Degradation and Mechanism of Action of PEA

PEA appears to exert its protective effect by decreasing the development of cerebral edema, down-regulating the inflammatory cascade, and limiting cellular necrosis and apoptosis [37,38]. PEA has been shown to inhibit peripheral inflammation and mast-cell degranulation, as well as to exert neuroprotective and antinociceptive effects in rats and mice [37,38]. The biosynthesis of PEA occurs through a common enzyme for the other NAEs, the selective phospholipase *N-*acyl-phosphatidyl-ethanolamine D (NAPE-PLD), starting from the hydrolysis of the precursor *N-*palmitoyl-phosphatidyl-ethanolamine (NAPE) [39]. The degradation of PEA occurs through the action of two different enzymes: the first selective for PEA, the amidase of *N-*acylethanolamine-hydrolysing acid (NAAA), while the second, the FAAH enzyme, instead deputy to the hydrolysis not only of PEA, but also of the other NAEs [31,40,41]. The anti-inflammatory, anti-hyperalgesic, and neuroprotective effects of PEA can occur in different ways.

The pleiotropic action of PEA involves multiple mechanisms [10]. Over the years, several PEA action mechanisms have been hypothesized, which do not exclude each other. The first, as previously mentioned, is the ALIA mechanism proposed by the research group of Rita Levi Montalcini, hence the name of this family of molecules, according to which PEA acts as a down-regulator of mast cell activation [42,43]. From the earliest studies, it was clear that PEA acted as a neutralizer of mastocyte degranulation, and to understand how they occurred, a second mechanism of action was thought to provide a direct link to the CB2 receptor. Initially it was thought that PEA could act as a CB2 receptor agonist [44], a theory that was later contradicted by the research group of Sugiura et al. [45], which showed that the latter had a low affinity for the CB2 receptor, thus explaining why PEA action on this receptor was not blocked by CB2 antagonists [46]. By virtue of this low affinity to bind CB2 receptor, a further mechanism that was mediated by the PEA is proposed through the direct activation of two other receptor targets: the peroxisome proliferator-activated receptors α (PPAR-α), belonging to the nuclear receptor superfamily [47] and the orphan G protein-coupled receptor (GPR55) [48]. Finally, a further theory has been advanced, which does not exclude the previous ones, according to which the PEA through a mechanism called “entourage effect” that can act indirectly on cannabinoid receptors, in particular towards CB2 receptor explaining its activation, despite the low affinity [49,50,51]. Through this entourage effect, PEA can indirectly activate the CB2 and CB1 receptors [50,52], probably by increasing the levels of AEA and 2-AG, for example by inhibiting the expression of FAAH, the enzyme that is responsible for their degradation [50], Similarly, PEA can indirectly activate type 1 vanilloid transient receptors (TRPV1) channel, an endogenous target of ALIAmides or increase AEA and 2-AG activation and desensitization of TRPV1 induced [49,51,52,53,54]. Recently, it has also been shown that PEA through PPAR-α receptors can increase the expression of CB2 receptors [55] and activate TRPV1 channels [56,57], Finally it was also proposed that PEA could improve the activation and desensitization of the AEA and 2-AG of the TRPV1 channels through an allosteric modulation of TRPV1 (entourage effect) [49,50,52]. See biosynthesis and degradation in Figure 1 and molecular target and mechanism of action of PEA in Figure 2.

## 6. PEA, Synergy with Natural Molecules with Antioxidant Properties

Among the molecules that the immune cells secrete in the tissues during the inflammatory processes, oxidizing agents play an important role in the innate immunity. However, oxidizing species in inflamed tissues produce a wide variety of harmful effects, the intensity of which contributes to pathological phenomena and broadens the intracellular response. The effects of highly reactive molecular species that were derived from oxygen and nitrogen to proteins, lipids, and nucleic acids cause tissue damage. Under conditions characterized by low-grade inflammation, the use of products able to minimize the toxic outcomes of reactive species of oxygen and nitrogen can contribute to the restoration of the tissue homeostasis. PEA lacks a direct antioxidant capacity to prevent the formation of free radicals, and to counteract the damage of DNA, lipids, and proteins. Furthermore, with its lipid structure and the large size of heterogeneous particles in the native state, PEA has limitations in terms of solubility and bioavailability. PEA has been micronized or ultra-micronized to overcome these problems. A major benefit of micro-crystallization is the enhanced rate of dissolution [58] and the rate of absorption of small drug particles is not influenced by the hydrodynamics in the gastro-intestinal tract an important factor in reducing variability of drug absorption when orally administered [58]. Impellizzeri et al. have tested these PEA formulations in carrageenan-induced inflammation in the rat paw extensively used in the development of anti-inflammatory drugs. Micronized and ultra-micronized PEA possessed superior pharmacological action against carrageenan-induced inflammatory pain, in contrast to the preparation of non-micronized PEA, which failed to show efficacy when orally given in this model [59]. The anti-inflammatory action of PEA combinated with an antioxidant could potentiate its pharmacological effects [31]. Among the natural molecules that are able to counteract the peroxidation processes there are several flavonoids, such as Luteolin, Polydatin, Quercetin, and Silymarin, which possess various pharmacological actions and therapeutic applications.

### 6.1. PEA and Luteolin

Luteolin is a flavone that is normally present in glycosylated form in foods commonly present in the diet, such as celery, carrots, salad, spinach, onion, oregano, rosemary, green pepper, chamomile, olive oil, and in plants, such as honeysuckle, chrysanthemum morifolium, and burdock [60,61,62,63]. It has been shown that luteolin has numerous biological effects: among these, the protective effect on the oxidative processes involving DNA in the presence of free radicals [64,65] and anti-inflammatory and anti-oxidant effects [66]. Luteolin is one of the most powerful and effective polyphenols that are capable of inhibiting the production of proinflammatory cytokines and nitric oxide induced by lipopolysaccharide endotoxin [67,68]. The anti-inflammatory activity of Luteolin includes the activation of anti-oxidative enzymes, suppression of the nuclear factor (NF)-B pathway, and inhibition of pro-inflammatory substances [69]. The pharmacodynamic properties of PEA and those of flavonoids luteolin and polydatin appear complementary, which suggests that, if administered in combination, they can counteract the two main conspirators of chronic diseases: low-grade inflammation and oxidative stress. Based upon the findings obtained with micronized/ultra-micronized PEA, these experiments were carried out while utilizing a co-ultra-micronized PEA/Luteolin composite (co-ultraPEALut, prepared in a 10:1 mass ratio), again making use of jet milling technology. In a chronic corticosterone mouse model of anxiety/depression has been shown for the first time that co-ultraPEALut exerts a significant antidepressant effect at low doses. Furthermore, this compound improves hippocampal neurogenesis and neuroplasticity, which are elements that influence cognition [70]. The search for molecules that participate in neuroprotection has become important, especially for protecting the nervous tissue from secondary neurodegenerative events. In this regard, it has been shown that co-ultraPEALut exerts a protective role in a model of spinal cord organotypic cultures and an in vivo model. In particular, this study showed that co-ultraPEALut significantly reduces inflammation and oxidative stress associated with damage. Furthermore, this compound improves motor function and histological alteration caused by spinal cord injury in mice [71]. In addition, another study suggests that co-ultraPEALut treatment inhibits autophagic cell death induced at the lesion site in in vivo model of SCI [72]. Neuroprotective propriety of co-ultraPEALut have also been demonstrated on secondary inflammatory process and autophagy involved in traumatic brain injury (TBI). In particular, co-ultraPEALut modulated apoptosis, the release of cytokine and reactive oxygen species, the activation of chymase, tryptase, and nitrotyrosine, and inhibited autophagy [73]. Unlike SCI and TBI, co-ultraPEALut has been shown to stimulate autophagy in in vivo and in in vitro Parkinson’s disease model (PD). Treatment with co-ultraPEALut is able to modulate both the neuroinflammatory process and the autophagic pathway that is involved in PD, actions, which may underlie its neuroprotective effect [74]. Inflammatory processes and oxidative stress also play an important role in Alzheimer’s disease (AD). Therefore, the effect of co-ultraPEALut was investigated in an organotypic model of AD, both in vitro and ex vivo [75]. In demyelinating diseases, such as multiple sclerosis, it has been shown that oligodendrocytes are not able to repair damage to themselves or to other cells. Therefore, given that co-ultraPEALut is able to improve spinal injuries and those that are caused by head trauma, research was carried out to evaluate the ability of the compound to promote the maturation of undifferentiated oligodendrocyte precursor cells (OPC). The study found that co-ultraPEALut promotes the morphological development of OPC and the total protein content without affecting proliferation [76]. Oxidative stress has a crucial role in the pathogenesis of inflammatory diseases, such as rheumatoid arthritis. In this regard, co-ultaPEALut was tested in the mouse model of collagen-induced arthritis to assess its ability to counteract the inflammatory process. The study showed that the compound had protective effects, not only on inflammation, but also on cartilage and bone in the tibiotarsal joints of animals [77]. Given the protective capabilities of co-ultraPEALut many clinical trials have been performed. For example, the protective effect and the synergy between PEA and Luteolin have been evaluated in patients suffering from stroke [78], or in patients with autism [79], or in patients who, following hip rupture, showed neuropsychiatric disorders, such as delirium [80]. See the co-ultraPEALut molecular targets in Table 1.

### 6.2. PEA and Polydatin

Polydatin (PLD) (3,5,4-dihydroxystylate-3-*O*-β-d-glucopiranoside) is a resveratrol glucoside in which the glycoside group is bound in the C-3 position as a substitute for a hydroxyl group. It is the predominant form in which resveratrol is present in nature. It has been suggested that glucoside can be hydrolyzed to resveratrol already in the duodenum or colon [85]. The biological properties of polydatin have been demonstrated in many studies and they include antioxidant activities, anti-inflammatory [86,87], and neuroprotective [88,89,90]; moreover, PLD improves microcirculation, inhibits platelet aggregation, asthmatic reactions, and showed microbial resistance and anti-cancer activities. These results led the researchers to evaluate the therapeutic effects of combination of PEA and PLD or the co-micronized compound m-(PEA/PLD). Today endometriosis (EM) is mostly considered to be a chronic inflammatory pathology with multifactorial aetiology, characterized by the implantation and growth of endometrium outside the uterine cavity [91]. Since EM touches 6–10% of the general female population and the frequency of the disease reaches 35–50% in women with pelvic pain, it is necessary to develop a therapy that is as individual and personalized as possible, while considering the problem in its entirety, in order to improve the quality of life of each patient [92]. Since the main features of the pathogenesis of EM include innate immunity, oxidative stress, and the presence of activated and degranulated mast cells inside the structures, which contribute to the development of pain and hyperalgesia [91,93,94], many researchers decided to use the m-(PEA/PLD) association. This association has proven to be effective and safe, as it has no side effects, both in pre-clinical and clinical studies [95,96,97,98,99,100]. The effect of PEA-PLD in patients with vestibulodynia that were subjected to transcutaneous electrical nerve stimulation (TENS) was also evaluated in a clinical study. This study showed that the combination PEA-PLD represents a significant additional treatment to TENS, as it is able to down-regulate the hyperactivity of mast cells that are responsible for the proliferation and propagation of the vestibular pain fibers and of hyperalgesia and allodynia [101]. The combination of PEA and PLD appears to be a promising adjunctive therapy in the treatment of primary dysmenorrhoea in adolescents and young women [102]. In a more recent study, it was shown that the m-(PEA/PLD) compound was able to reduce the inflammatory pain that is induced by the injection of carrageenan in the hind paws of rats. Furthermore, it was shown that m-(PEA/PLD), in addition to having peripheral action, also has an effect on downstream effectors that are associated with pain in the spinal cord [103]. The efficacy of the m-(PEA/PLD) compound was also evaluated in a benign testosterone-induced hyperplasia model (BPH). The results of this study reported that the co-micronized compound is able to reduce prostate weight and dihydrotestosterone production in BPH-induced rats. These effects are probably related to the anti-inflammatory and apoptotic effects of m-(PEA/PLD) [104]. Finally, since acute and chronic inflammatory responses are important risk factors for vascular remodeling processes, such as atherosclerosis, arteriosclerosis, and restenosis, it has been demonstrated in a model of vascular damage induced by complete artery ligation left carotid artery the protective effect of the combined treatment of PEA and PLD. In this study, it was clearly demonstrated that PEA/PLD reduces vascular damage, the expression of adhesion molecules, and attenuates the inflammatory process [105]. See the m-(PEA/PLD) molecular targets in Table 2.

### 6.3. PEA and Quercetin

Quercetin, which belongs to the group of flavonols, is the aglycon component of various glycosides, including rutin and quercitrin. It is one of the most common flavonoids since it can be isolated from numerous plant species including: horse chestnut, calendula, hawthorn, chamomile, hypericum and Ginkgo biloba. Foods that are particularly rich in quercetin are: the caper, red grapes and red wine, red onion, green tea, blueberry, apple, propolis, and celery. It is considered to be a natural inhibitor of various intracellular enzymes and for these properties it has been abundantly studied in the experimental oncological field, in the elucidation of the mechanisms of cell proliferation and carcinogenesis [106,107]. Quercetin is also a natural antioxidant. Among its most important functions are: restoring tocopherol (Vitamin E), after it has turned into a free radical (tocopheryl-radical), detoxifying the cell from the superoxide, and slowing down the production of nitric oxide during inflammation [108].

Currently, the new co-micronized compound consisting of PEA and quercetin has been tested in a few experimental models, including carrageenan-induced paw edema and osteoarthritis (OA) that is induced by sodium monoiodoacetate (MIA). OA represents one of the most frequently occurring painful conditions in both humans and small animals. Oxidative stress is considered to be an important etiologic factor in OA, and the antioxidant quercetin has been used with success as an adjunct in human and experimental arthritic diseases. In these studies, PEA-Q has been shown to reduce carrageenan-induced inflammatory responses and hyperalgesia. A reduction in mechanical allodynia with motor improvement and protection of the cartilage was also observed in animals that were treated with MIA. Currently, the translatability of these observations to canine and feline OA pain is currently under study [109]. See the PEA-Q molecular targets in Table 3.

### 6.4. PEA and Silymarin

*Silybum marianum* L., a member of *Carduus marianum* family, is an medicinal plant that has been used for centuries for the treatment of different diseases, such as liver, defensive liver against snake bite and insect stings, mushroom killing, and alcohol abuse [110]. Moreover, this herb has been used for centuries as a natural treatment for upper gastrointestinal tract and digestive problems, menstrual disorders, and varicose veins [111]. However, the very first usage of this herb was for its hepatoprotectant and antioxidant activities. Silymarin is the active component of this herb that exists in its fruit and seeds more than the other parts [112,113]. Silymarin effects have also been indicated in various illnesses of different organs, such as prostate, lungs, central nervous system, kidneys, pancreas, and skin [114]. According to pharmacological studies, silymarin has been accepted as a safe herbal product, since using the physiological doses of silymarin is not toxic. Silymarin has been combined with PEA in an animal model of kidney ischemia and reperfusion due to its antioxidant properties [115]. In this study, various indicators of renal dysfunction and tubular damage were evaluated, but also the inflammatory and apoptotic processes. The results that were obtained by the researchers show how the combined PEA-silymarin treatment was able to reduce histological damage, renal dysfunction, inflammation, and oxidative stress more than single substances. See the PEA-sylimarin molecular targets in Table 4.

### 6.5. PEA and Baicalein

Baicalein (5,6,7-trihydroxyflavone) is a flavone that was originally isolated from the roots of *Scutellaria baicalensis* and *Scutellaria lateriflora*. It is also reported in *Oroxylum indicum* (Indian trumpet flower) and Thyme. It is the aglycone of baicalin. Baicalein is one of the active ingredients of Sho-Saiko-To, a Chinese herbal supplement that is believed to enhance liver health. Various studies have proven that Baicalein show antioxidant properties via free radical scavenging capacity and/or enhancing antioxidant system [116]. Moreover, this traditional Chinese herb possesses cardiovascular protective [117], anti-thrombotic [118], anti-inflammatory [119], anti-apoptosis [120], anti-viral [121,122], and anti-cancer effects in vitro [123] and in vivo [124,125]. A combination of PEA and Baicalein has been recently studied in a myocardial I/R injury model [126]. In this study, treatment with PEA-Baicalein was shown to reduce myocardial tissue injury, neutrophil infiltration, markers for expression of mast cell activation, such as chymase and tryptase, and pro-inflammatory cytokine production (TNF-α and IL-1β). Furthermore, treatment with PEA-Baicalein reduced nitrotyrosine and PAR formation, inhibited NF-kB nuclear translocation and modulated apoptosis pathways. See the PEA-Baicalein molecular targets in Table 5.

## 7. Coffee Identification of a New Form of PEA

2-pentadecyl-2-oxazoline (PEA-OXA) is a new form of PEA that, besides having the classic anti-inflammatory, analgesic, and neuroprotective properties of the latter, also has the ability to modulate the catalytic activity of NAAA amidase, responsible for PEA degradation. Given that PEA is produced as needed and exerts pleiotropic effects on the cells that are involved in inflammation, modulating the activity of specific amidases involved in its degradation, NAAA and FAAH in particular, could represent a way to preserve PEA and help to maintain the cell homeostasis. In a recent study, the LC-MS technique was used to demonstrate the natural presence of PEA-OXA in coffee seeds and infusions [127]. This is the first work in which the natural presence of PEA-OXA is reported in the plant world. Pharmacological studies have been carried out to demonstrate the effects of the administration of PEA-OXA in different experimental models. The effect of PEA-OXA has been investigated in a carrageenan-induced rat inflammation model. PEA-OXA has visibly reduced histological damage, thermal hyperalgesia, inflammation, and oxidative stress. These effects also led to an increase in PEA at the level of the paw. This is probably due to the ability of PEA-OXA to modulate NAAA [128]. The modulation of intracellular NAAA by treatment with PEA-OXA was also evaluated in neuroinflammation models, such as TBI and SCI [129], in neurodegenerative Parkinson’s disease [130], in neuropathic pain after sciatic nerve crush [131], and in vascular dementia [132].

## 8. Conclusions

Compounds that are present in nature, assimilable through diets, can therefore play an important role in maintaining the well-being of each individual. The possibility through a healthy diet aimed at reinforcing our immune system, for example by enhancing the action of endogenous molecules, such as Aliamides, and, in particular, PEA, can be a valid alternative to the use of anti-inflammatory drugs, or why not, a possible therapy that works in synergy with the latter. PEA shows considerable versatility in counteracting inflammation, in all its formulations and derivatives, such as PEA-um or PEA-OXA. The possibility of using PEA in association with other natural antioxidant molecules, such as the flavonoids, such as Polydatin, Luteolin, Quercetin, or Silyrmarin, demonstrates that PEA is able to act effectively not only individually, but also and above all in synergy with other molecules. It can be concluded that all of these aspects reinforce the concept that sees PEA as an important endogenous balancer of the inflammatory process.

## Figures and Tables

**Figure 1 nutrients-11-02175-f001:**
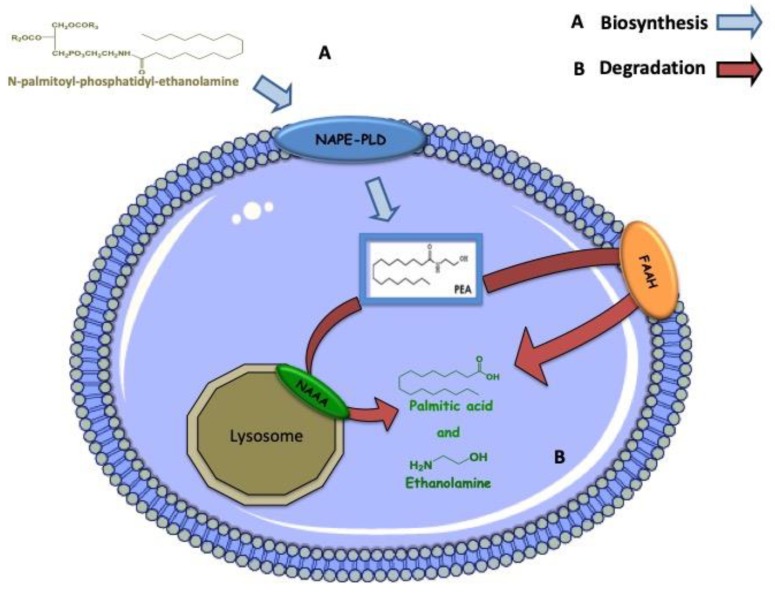
Metabolic pathways of palmitoylethanolamide (PEA). (**A**) PEA is biosynthesized from a membrane phospholipid, N-palmitoyl-phosphatidyl-ethanolamine (NPPE), through the direct hydrolysis by N-acyl phosphatidylethanolamine-specific phospholipase D (NAPE-PLD). (**B**) PEA can be then degraded to palmitic acid and ethanolamine by either FAAH or seletive enzime NAAA [10].

**Figure 2 nutrients-11-02175-f002:**
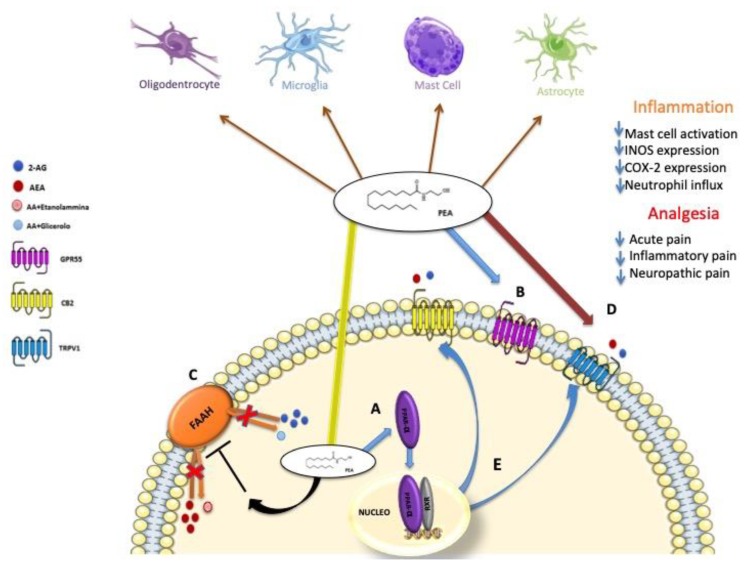
Molecular target and mechanism of action of PEA. (**A**)PEA can directly activate PPAR-α [47] or (**B**) GPR55 [48]. (**C**) PEA through the inhibition of the expression of FAAH, may increase the endogenous levels of AEA and 2-AG, which directly activate CB2 (or CB1) receptors and TRPV1 channels (entourage effect) [50,52]. (**D**) PEA, probably through an allosteric modulation of TRPV1 channels, potentiates the activation and desensitization by AEA and 2-AG of TRPV1 channels (entourage effect) [10]. (**E**) PEA may also activate TRPV1 channels via PPAR-α [56,57], or increase CB2 receptor expression via PPAR-α [55].

**Table 1 nutrients-11-02175-t001:** Co-ultraPEALut therapeutic actions and molecular targets: in vivo and in vitro study.

Type of Study	Molecular Targets	References
**Mouse model of Anxiety/Depressive**	BrdU ↑	[70]
	DCX ↑	
	BDNF ↑	
	Bax ↓	
	Bcl-2 ↑	
**Mouse model of Spinal Cord Injury**	Cox-2 ↓	[71,72,81]
	iNOS ↓	
	nNOS ↑	
	PPAR ↑	
	PPARβ/δ ↓	
	PPARγ ↓	
	Beclin-1 ↓	
	p62 ↓	
	MAP-LC3 ↓	
	mTOR ↑	
	p70S6K ↑	
	p-AKT ↑	
	BrdU ↑	
	DCX ↑	
	GFAP ↓	
	MAP-2 ↑	
	BDNF ↑	
	GDNF ↑	
	NGF ↑	
	NT-3 ↑	
**Mouse model of Rheumatoid Arthritis (CIA)**	Chymase ↓	[77]
	Tryptase ↓	
	Mast cells ↓	
	MIP-1α ↓	
	MIP-2 ↓	
	IL-1β ↓	
	IL-6 ↓	
	TNF-α ↓	
	MPO activity ↓	
	Nitrotyrosine ↓	
	MDA ↓	
**In vitro model of Alzheimer’s disease**	IκBa ↑	[75]
	NFkB p65 ↓	
	BDNF ↑	
	GDNF ↑	
	GFAP ↓	
	iNOS ↓	
	nNOS ↑	
	AIF ↓	
	Caspase-3 ↓	
	PARP-1 ↓	
**Mouse and in vitro model of Parkinson’s disease**	TH ↑	[74]
	DAT ↑	
	IκBa ↑	
	NFkB p65 ↓	
	GFAP ↓	
	TNF-α ↓	
	iNOS ↓	
	nNOS ↑	
	Cox-2 ↓	
	Bax ↓	
	Bad ↓	
	Bcl-2 ↑	
	mTOR ↓	
	p70S6K ↓	
	Beclin-1 ↑	
	p62 ↑	
	MAP-LC3 ↑	
**Demyelinating diseases (Maturation of Oligodendrocyte Precursor Cells)**	Cnr1 ↑	[76,82,83]
	Cnr2 ↑	
	Cat ↑	
	Cnp ↑	
	Hmgcr ↑	
	Idi1 ↑	
	Mki67 ↑	
	Mbp ↑	
	Plp1 ↑	
	Scd1 ↑	
	Sod2 ↑	
	Ugt8 ↑	
	PDGFR-α ↑	
**Mouse model of Traumatic Brain Injury**	IκBa ↑	[73]
	NFkB p65 ↓	
	TNF-α ↓	
	IL-1β ↓	
	GFAP ↓	
	Iba1 ↓	
	Chymase ↓	
	Tryptase ↓	
	GDNF ↑	
	iNOS ↓	
	pJNK ↓	
	Bax ↓	
	Caspase-3 ↓	
	mTOR ↑	
	p70S6K ↑	
	Beclin-1 ↓	
	p62 ↓	
	MAP-LC3 ↓	
**Rat model of Cerebral Ischemia (MCAO)**	GFAP ↓	[78]
	BDNF ↑	
	GDNF ↑	
	Mast cells ↓	
	Chymase ↓	
	Tryptase ↓	
	Bax ↓	
	Bcl-2 ↑	
**Mouse model of Autism**	IκBa ↑	[79]
	NFkB p65 ↓	
	iNOS ↓	
	GFAP ↓	
	TNF-α ↓	
	IL-1β ↓	
	Chymase ↓	
	Tryptase ↓	
	Bax ↓	
	Bcl-2 ↑	
	BrdU ↑	
	DCX ↑	
**Mouse model of Multiple Sclerosis (MS)**	SAA1 ↓	[84]
	TNF-α ↓	
	IL-1β ↓	
	IFN-γ ↓	
	TLR2 ↓	
	Fpr2 ↓	
	CD137 ↓	
	CD3-γ ↓	
	TCR-ζ chain ↓	
	CB_2_ ↓	
**Mouse model of Delirium**	Bax ↓	[80]
	Bcl-2 ↑	
	TNF- α ↓	
	IL-1β ↓	
	IκBa ↑	
	NFkB p65 ↓	
	Nrf-2 ↑	
	Mn-SOD ↑	
	GDNF ↑	

↑ : increase; ↓ : decrease.

**Table 2 nutrients-11-02175-t002:** m-(PEA/PLD) therapeutic actions and molecular targets: in vivo and in vitro study.

Type of Study	Molecular Targets	References
**Rat paw model of carrageenan-induced inflammation (in vivo and in vitro study)**	TNF-α ↓	[103]
	IL-6 ↓	
	IL-1β ↓	
	MPO ↓	
	IκBa ↑	
	NFkB p65 ↓	
	COX-2 ↓	
	iNOS ↓	
	Mn-SOD ↑	
**Mouse model of surgically-induced Endometriosis**	MMP9 ↓	[99]
	Mast Cell ↓	
	NGF ↓	
	VEGF ↓	
	ICAM-1 ↓	
	MPO ↓	
	IκBa ↑	
	NFkB p65 ↓	
	Nitrotyrosine ↓	
	PAR ↓	
**Rat model of Benign Prostatic Hyperplasia**	PGE2 ↓	[104]
	DHT ↓	
	5α-reductase 1 ↓	
	5α-reductase 2 ↓	
	IκBa ↑	
	NFkB p65 ↓	
	iNOS ↓	
	COX-2 ↓	
	Nrf-2 ↑	
	HO-1 ↑	
	Mn-SOD ↑	
**Mouse model of Vascular Injury**	ICAM-1 ↓	[105]
	V-CAM ↓	
	TNF- α ↓	
	IL-1β ↓	
	iNOS ↓	
	IκBa ↑	
	NFkB p65 ↓	
	Bax ↓	
	FAS-Ligand ↓	
	α-sma ↑	
	MCP-1 ↑	
	BrdU ↑	

PEA, Palmitoylethanolamide. ↑ : increase; ↓ : decrease.

**Table 3 nutrients-11-02175-t003:** 2-pentadecyl-Q (PEA-Q) therapeutic actions and molecular targets: in vivo study.

Type of Study	Molecular Targets	Reference
**Rat paw model of carrageenan-induced inflammation and osteoarthritic pain model**	TNF-α ↓	[109]
	IL-1β ↓	
	MPO ↓	
	NGF ↓	
	MMP-1 ↓	
	MMP-3 ↓	
	MMP-9 ↓	

↑ : increase; ↓ : decrease.

**Table 4 nutrients-11-02175-t004:** PEA-sylimarin therapeutic actions and molecular targets: in vivo study.

Type of Study	Molecular Targets	Reference
**Mouse model of Kidney Ischemia and Reperfusion**	MPO ↓	[115]
	TNF-α ↓	
	IL-1β ↓	
	Nitrite/Nitrate ↓	
	Superoxide ↓	
	CuZn SOD ↑	
	Mn-SOD ↑	
	Catalase ↑	
	Nitrotyrosine ↓	
	PAR ↓	
	MDA ↓	
	Chymase ↓	
	ICAM-1 ↓	
	p-selectin ↓	
	IκBa ↑	
	NFkB p65 ↓	
	Bax ↓	
	Bcl-2 ↑	

↑ : increase; ↓ : decrease.

**Table 5 nutrients-11-02175-t005:** PEA-Baicalein therapeutic actions and molecular targets: in vivo study.

Type of Study	Molecular Targets	Reference
**Rat model of myocardial I/R injury**	MPO ↓	[126]
	Mast Cell ↓	
	Chymase ↓	
	Tryptase ↓	
	IκBa ↑	
	NFkB p65 ↓	
	TNF-α ↓	
	IL-1β ↓	
	Bax ↓	
	Bcl-2 ↑	

↑ : increase; ↓ : decrease.

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
