# Peer review of "Therapeutic Efficacy of Palmitoylethanolamide and Its New Formulations in Synergy with Different Antioxidant Molecules Present in Diets"

_nutrients, 2019, doi:10.3390/nu11092175_

Round 1

Reviewer 1 Report

This manuscript is giving clear summary of anti-inflammatory properties of ALIAmide in diet which will give a new perspective on related issues.

In the title, “analgesic” can be removed. Hard to find related summary and description in manuscript.

Most of the abbreviations should be accompanied by their full name at the first appearance in manuscript and a list of abbreviation should be added in manuscript.

Several points are indicated in attached file.

Author Response

Response to Reviewer 1 Comments

Point 1: This manuscript is giving clear summary of anti-inflammatory properties of ALIAmide in diet which will give a new perspective on related issues.

 Response 1: Thank you for the comment.

Point 2: In the title, “analgesic” can be removed. Hard to find related summary and description in manuscript.

Response 2: We provide to remove on the title analgesic as indicated.

 Point 3: Most of the abbreviations should be accompanied by their full name at the first appearance in manuscript and a list of abbreviation should be added in manuscript.

Response 3: We provide to add a list of abbreviation after keyword and abstract in the manuscript

Point 4: Several points are indicated in attached file.

Response 4: Thank you. We provide to modify in the manuscript some point.

Reviewer 2 Report

This review highlights the therapeutic efficacy of palmitoylethanolamide (PEA) and its new formulations in synergy with different antioxidant molecules. Authors are experienced in this area, having published different articles on the beneficial effects of PEA in vascular dementia, regional pain syndrome, cerebral ischemia, spinal cord and brain injury, arthritis or drug induced neurotoxicity. They have also published different reviews on PEN as a novel therapeutic strategy (Esposito and Cuzzocrea 2013, Skaper et al, 2015). The present article aims to expand previous reviews to cover the whole area of PEN beneficial effects.

Comments:

As correctly indicated in the final sentence of the abstract, the manuscript mainly reviews PEN synergy with other molecules. This should be made clear in the title of the article and at the end of section 1, replacing the present unspecific sentence “in this review we will see how inflammation can be prevented or combated by diets or compounds present in some food”, and mentioning the nature of the molecules synergizing with PEA (antioxidant flavonoids and stilbenes).

The review does not only cover anti-inflammatory effects of PEA but also other mechanisms responsible for its therapeutic efficacy (i.e. analgesic action), and this should be considered when modifying the title of the article.

Review is highly descriptive. More emphasis should be made on PEA mechanisms of action and, if possible, identified molecular targets should be mentioned when describing specific studies.

Synergistic effects of PEA with flavonoids and stilbenes is probably the result of acting on  to complementary phenomena, inflammation and the formation of ROS, that independently feed cell death (see Petrosino and Di Marzo, 2017). This is worthy to comment in the manuscript.

In order to make the manuscript more comprehensive to readers, a figure representing PEA metabolism pathways/molecular targets and a schematic illustration of PEA therapeutic actions should be added.

Previous reviews by the same authors should be mentioned (i.e. Esposito and Cuzzocrea, MiniRev Med Chem 2013; Esposito and Cuzzocrea, CNS Neurol Disorder Drug Targets 2013).

Recent research by different authors on the synergy of PEA with luteolin (Contarini et al, J Neuroinflammation 2019; Skaper et al, Mol Neurrobiol, 2018) or baycalein (D’amico et al, Phytomedicine, 2019) is lacking and should be cited.

Minor points:

-  According to the journal Instructions for authors, journal titles should be abbreviated in the final list of references.

-  There are different typographical mistakes that should be corrected (i.e. “Palmytoethanolamide” in the abstract, “With” in section 6 line 10, “modify” in the tile of section 4, “inflamaging” in section 1 line 3,and others).

Author Response

Response to Reviewer 2 Comments

Point 1: As correctly indicated in the final sentence of the abstract, the manuscript mainly reviews PEN synergy with other molecules. This should be made clear in the title of the article and at the end of section 1, replacing the present unspecific sentence “in this review we will see how inflammation can be prevented or combated by diets or compounds present in some food”, and mentioning the nature of the molecules synergizing with PEA (antioxidant flavonoids and stilbenes).

Response 1: Thank you for the comment. We provide to modify the title of the manuscript mentioning the synergy with the antioxidant molecules..

Point 2: The review does not only cover anti-inflammatory effects of PEA but also other mechanisms responsible for its therapeutic efficacy (i.e. analgesic action), and this should be considered when modifying the title of the article.

Response 2: Thank you for the comment. We provide to modify the title of the manuscript.

Point 3: Review is highly descriptive. More emphasis should be made on PEA mechanisms of action and, if possible, identified molecular targets should be mentioned when describing specific studies.

Response 3: Thank you for the comment. We provide to extend the session relating to the PEA mechanisms of action by adding other references where possible

Point 4: Synergistic effects of PEA with flavonoids and stilbenes is probably the result of acting on  to complementary phenomena, inflammation and the formation of ROS, that independently feed cell death (see Petrosino and Di Marzo, 2017). This is worthy to comment in the manuscript.

Response 4: Thank you for the comment. We provide to mentioning this part in the manuscript in the session relative to the synergy with the flavonoids

Point 5: In order to make the manuscript more comprehensive to readers, a figure representing PEA metabolism pathways/molecular targets and a schematic illustration of PEA therapeutic actions should be added.

Response 5: Thank you for the comment. We provide to add two figures, the first for the metabolism of PEA, and the second relating the molecular target and the potential mechanisms of action of PEA.

Point 6: Previous reviews by the same authors should be mentioned (i.e. Esposito and Cuzzocrea, MiniRev Med Chem 2013; Esposito and Cuzzocrea, CNS Neurol Disorder Drug Targets 2013).

Response 6: Thank you for the comment. We provide to mentioning these studies in the manuscript.

Point 7: Recent research by different authors on the synergy of PEA with luteolin (Contarini et al, J Neuroinflammation 2019; Skaper et al, Mol Neurrobiol, 2018) or baycalein (D’amico et al, Phytomedicine, 2019) is lacking and should be cited.

Response 7: Thank you for the comment. We provide to mentioning these studies in the manuscript and also add a session with the synergy between PEA and Baycalein-

Minor points:

Point 8: According to the journal Instructions for authors, journal titles should be abbreviated in the final list of references.

Response 8: Thank you for the comment. We provide to modify the reference list

Point 9: There are different typographical mistakes that should be corrected (i.e. “Palmytoethanolamide” in the abstract, “With” in section 6 line 10, “modify” in the tile of section 4, “inflamaging” in section 1 line 3,and others).

Response 9: Thank you for the comment. We provide to modify the typographical mistakes.

Round 2

Reviewer 2 Report

Most  comments and suggestions have been adequately answered.

However,  previous reviews by the same authors should still be cited (i.e. Esposito and Cuzzocrea. MiniRev Med Chem 2013, ESposito and Cuzzocrea. CNS Neurol Disorder Drug Target 2013).

Moreover, there are ortographical and syntax mistakes, specially in the new added text; proof reading is required. Some examples: Page 2. infllamming (should be inflammaging). Page 2 ...a good diet meets us by giving us excellent alternatives to the traditional drug (improve sentence).

Author Response

Response to Reviewer 2 Round 2 Comments

Point 1: However,  previous reviews by the same authors should still be cited (i.e. Esposito and Cuzzocrea. MiniRev Med Chem 2013, ESposito and Cuzzocrea. CNS Neurol Disorder Drug Target 2013).

Response 1: Thank you for the comment. We provide to cited to paragraph 5 on page 5 the two previous reviews

Point 2: Moreover, there are ortographical and syntax mistakes, specially in the new added text; proof reading is required. Some examples: Page 2. infllamming (should be inflammaging). Page 2 ...a good diet meets us by giving us excellent alternatives to the traditional drug (improve sentence)

Response 2: Thank you for the comment We provide to correct orthographical mistake and modify some sentence.

This manuscript is a resubmission of an earlier submission. The following is a list of the peer review reports and author responses from that submission.